# A Comprehensive Spatio-Temporal Model for Subway Passenger Flow Prediction

Zhihao Zhang [1,2], Yong Han [1,2,*], Tongxin Peng [1,2], Zhenxin Li [1,2] and Ge Chen [1,2]

1   Faculty of Information Science and Engineering, Ocean University of China, No. 238, Songling Road, Qingdao 266100, China; zhangzhihao3974@stu.ouc.edu.cn (Z.Z.); tongxinp@stu.ouc.edu.cn (T.P.); lizhenxin@stu.ouc.edu.cn (Z.L.); gechen@ouc.edu.cn (G.C.)
2   Laboratory for Regional Oceanography and Numerical Modeling, Qingdao National Laboratory for Marine Science and Technology, No. 1, Wenhai Road, Qingdao 266237, China
*   Correspondence: yonghan@ouc.edu.cn

**Abstract:** Accurate subway passenger flow prediction is crucial to operation management and line scheduling. It can also promote the construction of intelligent transportation systems (ITS). Due to the complex spatial features and time-varying traffic patterns of subway networks, the prediction task is still challenging. Thus, a hybrid neural network model, GCTN (graph convolutional and comprehensive temporal neural network), is proposed. The model combines the Transformer network and long short-term memory (LSTM) network to capture the global and local temporal dependency. Besides, it uses a graph convolutional network (GCN) to capture the spatial features of the subway network. For the sake of the stability and accuracy for long-term passenger flow prediction, we enhance the influence of the station itself and the global station and combine the convolutional neural networks (CNN) and Transformer. The model is verified by the passenger flow data of the Shanghai Subway. Compared with some typical data-driven methods, the results show that the proposed model improves the prediction accuracy in different time intervals and exhibits superiority in prediction stability and robustness. Besides, the model has a better performance in the peak value and the period when passenger flow changes quickly.

**Keywords:** deep learning; passenger flow prediction; hybrid model; spatio-temporal dependency

## 1. Introduction

By 2050, about 7 in 10 people will live in cities [1]. With the increasing urban population, urban transportation is expanding rapidly, which poses new challenges to the sustainable development of cities. Compared with private vehicles, urban rail transit can reduce transport-related energy consumption, travel costs, traffic congestion, and environmental pollution. Meanwhile, studies show that the growth of car ownership is relatively slow in cities with high urban rail transit intensity [2]. Therefore, the subway, bus, and other public transport facilities play a more important role in realizing the sustainable development of cities. Among them, the subway transportation system is an important measure to eliminate the shackles of urban traffic, alleviate urban traffic congestion, and build an urban three-dimensional transportation system due to a large traffic flow, fast operation speed, and small floor area [3]. Besides, the subway emits fewer pollutants and saves more energy, which can help promote the green and healthy development of the city.

Therefore, a timely and effective subway transportation system is very crucial. To avoid traffic congestion or traffic paralysis caused by insufficient subway resources, we can utilize passenger flow prediction to realize an effective allocation of traffic resources. Meanwhile, with the wide adoption of location-aware technology, urban spatiotemporal data (such as bus card data, subway card swiping data, mobile GPS, etc.) becomes abundant [4]. Therefore, the passenger flow can be extracted, which is helpful in studying human activity patterns. Intelligent passenger flow prediction based on big data can also

facilitate the development of the intelligent transportation system (ITS) [5]. In summary, it is achievable and imperative to develop a deep learning framework for subway passenger flow prediction.

Due to the great practical value and challenge of subway passenger flow prediction, researchers devote their energy more to the research of traffic flow prediction. The existing passenger flow prediction methods can be mainly classified into statistical methods, machine learning (ML) methods, and deep learning (DL) methods.

The statistical methods determine the parameters through the processing of the original data and realize the traffic prediction according to the regression function [6], such as the autoregressive integral moving average model (ARIMA) [7,8] and its variations [9–11], and the Kalman filter model [12–14]. They can easily calculate and capture the linear characteristics of the data. However, they rely on stationary assumptions and can be largely influenced by the fluctuant traffic data. In addition, they are difficult to reflect the non-linear and complex characteristics of the traffic flow.

The machine learning methods can obtain the non-linear features and statistical laws of the traffic data through sufficient historical observations, which can handle the problem of the statistical methods, such as the support vector regression (SVR) [15–17], k-nearest neighbor model (KNN) [18–20], Bayesian model [21,22], support vector machine (SVM) [23–25], and fuzzy logic model [26,27]. They improve the accuracy of the subway flow prediction, but it is difficult for them to achieve good results in complex networks with numerous nodes. They mostly rely on the complex manual feature engineering, which results in the lack of robustness to model massive data, and they are incapable of processing raw spatiotemporal data [28]. Therefore, it is difficult for the machine learning methods to obtain the best prediction results based on the abundant spatiotemporal data.

The deep learning methods can automatically establish feature engineering and improve the feature expression. Moreover, deep learning models have advantages in capturing non-linear and complex patterns, which can help them to get more accurate results. At present, the deep learning methods commonly used in spatiotemporal traffic flow prediction, such as the recurrent neural network (RNN) [29–31], the convolutional neural network (CNN) [32], the graph neural network (GNN) [33,34], etc. Traffic flow prediction essentially depends on historical observations. Therefore, temporal dependency is an indispensable part. However, some deep learning models [35,36] only consider the temporal dependency of passenger flow and ignore the spatial dependency. In this way, the traffic prediction is divorced from spatial factors, such as roads and stations. By integrating the spatial dependency, we can further improve the accuracy of the model. Therefore, aiming at the shortcomings of the single model in the passenger flow prediction, some studies [37–40] introduce the CNN to model spatial dependency, combining it with the RNN [41] model and its variant models [42,43]. However, due to the non-Euclidean and time-varying characteristics of the subway network, it is difficult for the CNN to describe the complex spatial topological relationship. Therefore, some deep learning models [44–46] introduce the graph convolutional neural network (GCN) [47,48] to improve the capture of spatiotemporal features in the passenger flow. At the same time, the RNN models have limitations for capturing the temporal dependency. The attention model [49] can capture the global and dynamic spatiotemporal characteristics, which is helpful for the prediction. Some deep learning models [50–53] introduce the attention model into the field of traffic flow prediction.

Much progress has been made in this task. However, some knowledge gaps still exist. Statistical methods are difficult to capture complex characteristics. Machine learning methods heavily depend on the manually designed characteristics. As for the deep learning methods, the existing models still have the following gaps:

(1) Most methods based on the GCN ignore the improvements to the adjacent matrix in subway passenger flow prediction. Firstly, they ignore the spatial influence of the import and export of subway stations. Secondly, they ignore the influence of global stations on specific stations.

(2)　Most methods are based on a single model to capture the temporal dependency, such as the RNN model and its variations, or the Transformer [54] model. However, these models still have limitations in capturing all the temporal characteristics. The RNN model and its variations only focus on the capture of local temporal characteristics, while the Transformer model only focuses on the capture of global temporal characteristics.

(3)　Traffic prediction is commonly classified into two scales, which are short-term ($\leq$30 min) and long-term ($\geq$30 min) [44,53]. At present, the subway passenger flow prediction is mostly a short-term prediction. However, the long-term flow prediction is also very important, which can provide more sufficient preparation for operation scheduling.

In summary, to solve the above problems and better predict the subway passenger flow, the GCTN is proposed. It can comprehensively model the local and global spatiotemporal dependency. Specifically, the original characteristics are obtained through the diffusion of passenger flow features by a CNN. We believe temporal feature modeling based entirely on captured spatial features will ignore part of the passenger flow characteristics. Therefore, the original characteristics and the characteristics captured by GCN are fused as the basis to capture the temporal characteristics. Later, the features containing original and spatial features were input into the LSTM and Transformer. The LSTM can capture the local temporal features, and the Transformer can capture the global temporal features. We integrate the local and global temporal characteristics to obtain comprehensive temporal characteristics. Then, to model synthesize the spatiotemporal dependency, and balance the spatial and temporal influence, we further fused the spatial characteristics into the comprehensive temporal characteristics. Therefore, we think the model can improve the accuracy and stability of long-term prediction.

After exploring the subway passenger flow data, we divide the subway passenger flow data into three patterns: close, daily, and weekly. This paper tries several fusion methods to explore the influence of the nearest neighbor and periodic segments. Compared with the current subway passenger flow prediction model, the proposed hybrid model in this paper has the following advantages:

1.　Through the improvements in the adjacent matrix and GCN, the expression of the spatial structure in the subway network is further obvious. Besides, we can better describe that station characteristic and the influence of global stations, which improves the capture of spatial characteristics.

2.　Modeling the spatiotemporal dependency comprehensively. We design a spatiotemporal block structure, and the objective is that seamlessly model the characteristics extracted by the network.

3.　An improvement is made in the Transformer model. The CNN is added to extract and fuse the intermediate features, which can help better analyze the subway time-series data.

4.　The motivation behind the GCTN is accurately predicting long-term subway passenger flow. Passenger flow prediction can help subway dispatching, and it can assist citizens in planning routes and scheduling their time. Moreover, it can help reduce traffic pressure and construct a healthy, green, and sustainable city.

The rest of this paper is organized as follows. In Section 2, we describe the mathematical expression of the prediction problem. Then, we introduce the model architecture in Section 3. In Section 4, we show the results in case analysis and discuss the influence factors of prediction results. Finally, in Section 5, we summarize the results and limitations of this paper and explore the future research direction.

## 2. Problem Definition

The subway passenger flow prediction is a problem of spatiotemporal prediction. First of all, the spatial structure in the subway can be expressed as a graph structure, G = (V, E, A). V is the set of N nodes representing the subway stations. E is the edge between subway stations. In addition, A ∈ $R^{N \times N}$ is the adjacent matrix based on the connectivity and Euclidean distance between subway stations. Moreover, subway passenger flow is not only affected by the spatial structure of the subway network, but also by the passenger flow in historical periods. Therefore, the prediction problem can be expressed by the following Equation (1):

$$Target = F_{prediction}\ (X,\ G,\ W), \tag{1}$$

where Target ∈ $R^{N \times T' \times 2}$ is the predicted subway passenger flow of *N* stations in time *T'*. The X ∈ $R^{N \times T \times 2}$ is the historical subway passenger flow on *N* stations in time *T*. The Target and X both include inflow volume and outflow volume. $F_{prediction}$ is the deep learning model. G is the graph structure of the subway network, and W is the learnable parameter.

In this paper, historical subway passenger flow is divided into three patterns (close, daily, and weekly). The close pattern represents the recent time, while the daily and weekly patterns represent the historical situation of passenger flow at the target time on different days or weeks. For example, if we predict the passenger flow in the next time slice. The close patterns of the target data are the previous few time slices before the target time. The daily patterns are the time slices at the same time in the previous few days. In addition, the weekly patterns are the time slices at the same time in the previous few weeks. Different patterns assist in studying the influence of periodic time slices on subway passenger flow at the target time. The prediction process is shown in Figure 1.

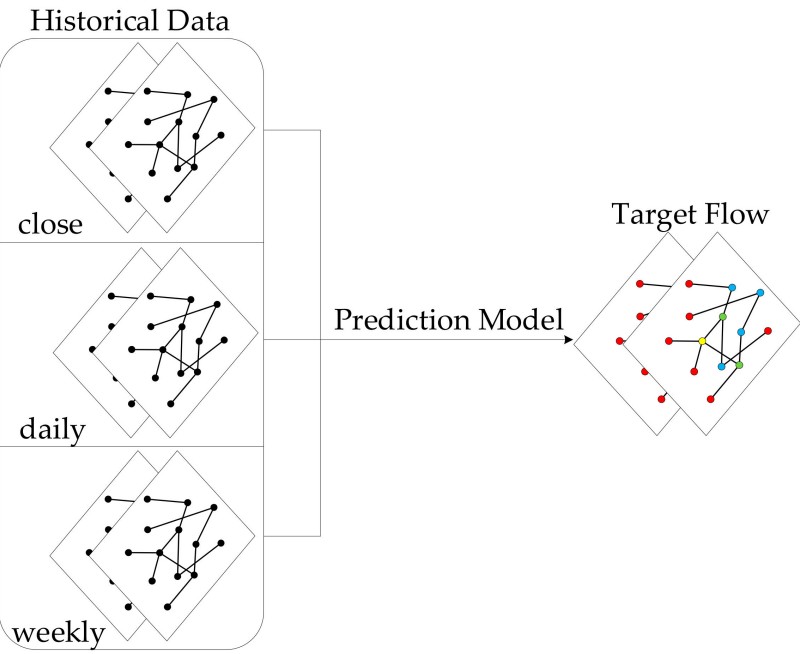

**Figure 1.** Schematic diagram of the prediction process.

This paper builds a hybrid neural network model GCTN (graph convolutional and comprehensive temporal neural network) to solve the problem of prediction in subway passenger flow. In addition to capturing the spatiotemporal dependency, the proposed model considers the correlation between stations with different time steps to strengthen the long-term passenger flow prediction.

At present, the commonly used long-term prediction method is autoregressive multi-step prediction. However, the predicted error will spread and gradually accumulate in the

autoregressive multi-step prediction. Therefore, we directly model long-term temporal dependency from historical data, which is used to predict multiple target flows. Meanwhile, to avoid error diffusion, we carry out parallel training through the joint loss of multiple time steps, which may achieve a more accurate multi-step prediction. The model is tested and verified by the passenger flow data in time intervals (TIs) of 10 min, 20 min, and 30 min.

## 3. Model Development

In this section, a graph-based neural network is employed to deal with the topological features and realize a long-term subway passenger flow prediction. We will present a brief review of the architecture of the GCTN model and the related mechanism.

### 3.1. Overall Architecture

As depicted in Figure 2, we designed a spatiotemporal block, which contains GCN, bidirectional long short-term memory (Bi-LSTM) network and Transformer models. The GCN model captures spatial features of the subway network by automatically learning the feature information and the structural information of stations. The path of input and output components in the Bi-LSTM model will influence the learning about the temporal dependencies. The longer the path, the harder it is to learn long-range dependencies effectively [54]. Therefore, the Bi-LSTM model is used to capture local temporal features. The Transformer model, based on the self-attention mechanism, can model the global context to capture long-range dependencies. Therefore, the Transformer model is used to capture global temporal features. To comprehensively model spatiotemporal dependencies, we design a spatiotemporal block. It can extract and fuse the spatiotemporal features. Moreover, the depth of the model can be increased by stacking spatiotemporal blocks, which is helpful in modeling the deep spatiotemporal dependencies. Then, the characteristics processed by spatiotemporal blocks are aggregated into predicted output values through two $1 \times 1$ convolutional layers. In Figure 2, the $X_{input}$ is the original passenger flow feature for characteristic diffusion through a CNN layer. The $X_{output}$ is the passenger flow feature after feature extraction and fusion through a spatio-temporal block. The $Y_{output}$ is similar to $X_{output}$, and when the number of spatio-temporal blocks is one, $Y_{output} = X_{output}$. The $Y_{prediction}$ is the predicted subway passenger flow.

### 3.2. Modeling Spatial Dependency

A crucial problem in subway passenger flow prediction is the complex spatial dependency of the subway network. The CNN is usually used to process Euclidean-structure data and obtain local spatial characteristics. However, the subway network is a graph structure in space, and it contains complex topological features. Meanwhile, it is difficult for the CNN to process non-Euclidean-structure data. Consequently, the CNN cannot effectively extract the topological spatial features in subway networks. GCN can simultaneously make an end-to-end prediction of node features and structure information. It can aggregate the information of adjacent nodes to obtain the characteristics of nodes in graph networks, which is based on the learnable weight and predefined graph. In addition, it is suitable for GCN to capture the spatial features of nodes and graphs with arbitrary topology. Therefore, GCN can capture spatial features in subway networks. Through GCN, the spatial relationship between subway network stations can be established. In addition, the accurate expression of spatial features is realized. How the GCN model captures the spatial characteristics between stations can be shown in Figure 3 and Equation (4).

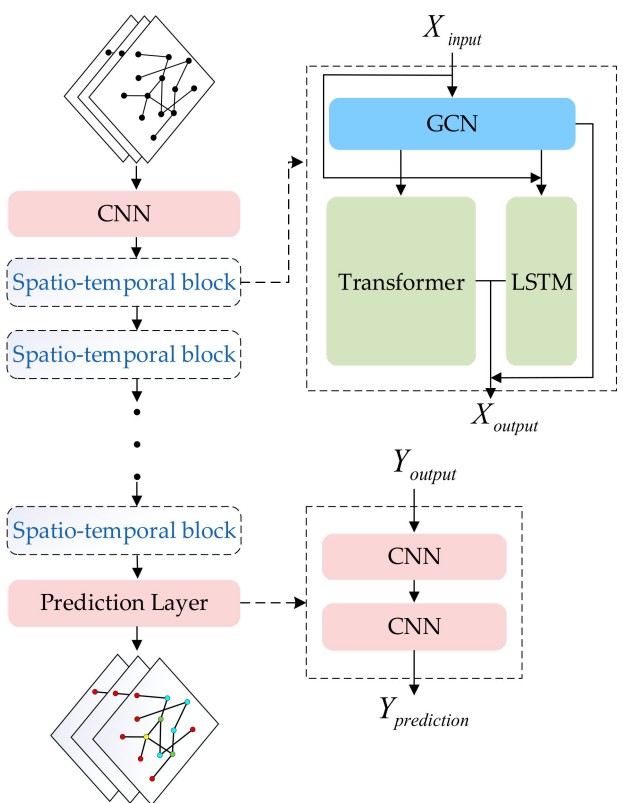

**Figure 2.** The overall architecture of the model.

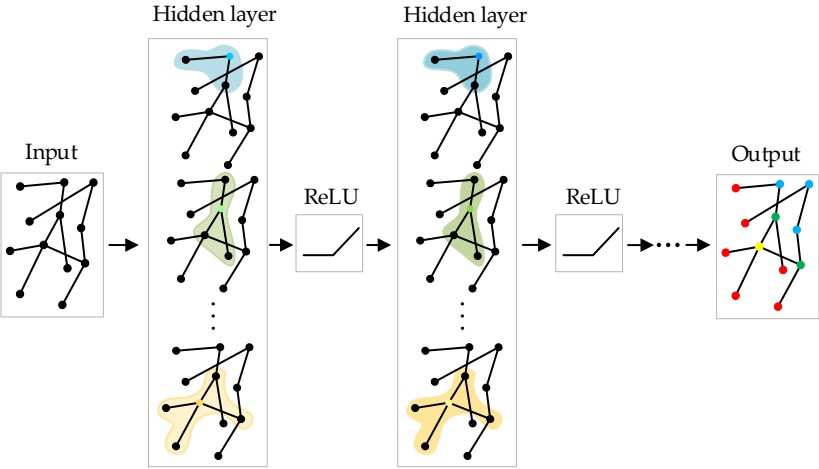

**Figure 3.** The diagram of capturing node features through GCN.

To enhance the prediction accuracy and stability, we improve the traditional adjacent matrix. There are many imports and exports in a subway station, and the different imports and exports may not be close to each other. People tend to take the import or export route closer to their location or destination. In addition, different numbers of imports and exports in subway stations may have different effects on subway passenger flow. Therefore, the spatial influence of imports and exports on the station is worth considering. We take the import and export in the subway station as a part of the feature to construct the adjacent matrix. In other words, the identity matrix is replaced by the diagonal matrix, which combines the import and export in the subway adjacent matrix. Therefore, the characteristics of the station itself are enhanced. Firstly, we standardize the number of imports and exports according to the maximum and minimum values. Secondly, we

construct a diagonal matrix based entirely on the normalized number of imports and exports. We construct an adjacent matrix representing the connectivity and Euclidean distance between stations. After that, we combine the adjacent matrix, the diagonal matrix, and the identity matrix. Lastly, the self-loops of the adjacent matrix are constructed, which can be shown in Equation (2).

$$\widetilde{A} = A + I_n + I_e,$$

(2)

where $\widetilde{A}$ represents the adjacent matrix with self-loops. $A$ represents the adjacent matrix established based on connectivity and Euclidean distance. $I_n$ is an identity matrix, and $I_e$ is a diagonal matrix constructed of the normalized quantity of import and export.

In GCN, the information of adjacent nodes is aggregated with the predefined adjacent matrix. After that, the spatial characteristics between nodes are obtained with the learned weight. The GCN model can be established through the stack of graph convolutional layers. We learn the structure-aware node features from the Chebyshev polynomial approximation, which can model spatial dependency. The historical data are $X_{input} \in R^{T \times N \times F}$, where $T$ represents the time steps of input features, $N$ represents the number of nodes, and $F$ represents the number of input features. The captured spatial characteristics are $X_s \in R^{T \times N \times F'}$, where $F'$ represents the number of output features, which can be represented by the following Equation (3):

$$X_s = \sigma \left( \sum_{k=0}^{K} \theta_k T_k \left( \widetilde{L} \right) X_{input} \right),$$

(3)

where $\sigma$ is a non-linear activation function, and we use ReLu() function. $k$ is the order of Chebyshev polynomials $T_k$, and we can regard it as the size of receptive field on convolution kernel. $\theta_k$ is the learnable weight parameter. The scaled Laplacian matrix is $\widetilde{L} = 2L/\lambda_{max} - I_n$ for Chebyshev polynomial, where the normalized Laplacian matrix $L$ is defined by $L = I_n - D^{-1/2} \widetilde{A} D^{-1/2}$, and $\lambda_{max}$ is the maximum eigenvalues of matrix $L$. $D$ is the degree matrix. In this paper, we use the diagonal elements synthesized by $\widetilde{A}$ to represent the degree matrix, namely $D_{ii} = \sum_j \widetilde{A}_{ij}$, $i, j = 1, \ldots, N$.

The passenger flow of adjacent stations will affect those of the current station. Meanwhile, the influence of the global stations on the specific station and its adjacent stations is worth considering. We added the dot product attention mechanism into the Chebyshev graph convolutional network. The scaled Laplacian operator is used in dot product operation to globalize the spatial features between stations, which may enhance the influence of global stations on a specific station and its adjacent stations.

Since there is inflow volume and outflow volume in subway passenger flow prediction, we conduct high-dimensional diffusion of the flow features through the CNN. The convolutional layer of the Chebyshev diagram needs to be modified. The modification is shown in Equation (4):

$$X_s = \sigma \left( \sum_{i=1}^{f} \sum_{k=0}^{K} \theta_{k,i} T_k \left( S \left( \widetilde{L} \right) \right) X_{input} \right)$$

(4)

where, $f$ is the number of feature channels. $S$ represents the dot product attention operation in the Laplacian operator.

### 3.3. Modeling Temporal Dependency

The important factors affecting temporal dependency include the proximity, trend, and periodicity of time slices. The proximity refers to the influence of the adjacent time intervals, the trend refers to the overall trend over some time, and the periodicity is the influence of a longer time [28]. In this paper, we try a variety of combined methods based on the close, daily, and weekly patterns. We capture temporal features by using the combination of time patterns and spatial features. We input the features into the LSTM and Transformer, respectively, and the objective is to capture the local and global temporal

features. In addition, we then fuse the two kinds of temporal characteristics to obtain the final comprehensive temporal characteristics.

Compared with the traditional RNN network, the LSTM solves the gradient disappearance problem caused by the gradual reduction in the gradient backpropagation process, and it can construct a deeper neural network [42]. At the same time, compared with the traditional RNN and GRU, the LSTM has fewer errors in flow prediction [30].

The subway flow of a specific time step is affected by the past information as well as the future information. Therefore, we use the Bi-LSTM network to capture the influence of passenger flow at the front and back time steps on the current time step. It may enhance the local temporal characteristics and make the local temporal features more significant. The Bi-LSTM network is a combination of the forward LSTM and the backward LSTM, and its main components include the input layer, the recurrent hidden layer, and the output layer. The schematic diagram of the Bi-LSTM is shown in Figure 4.

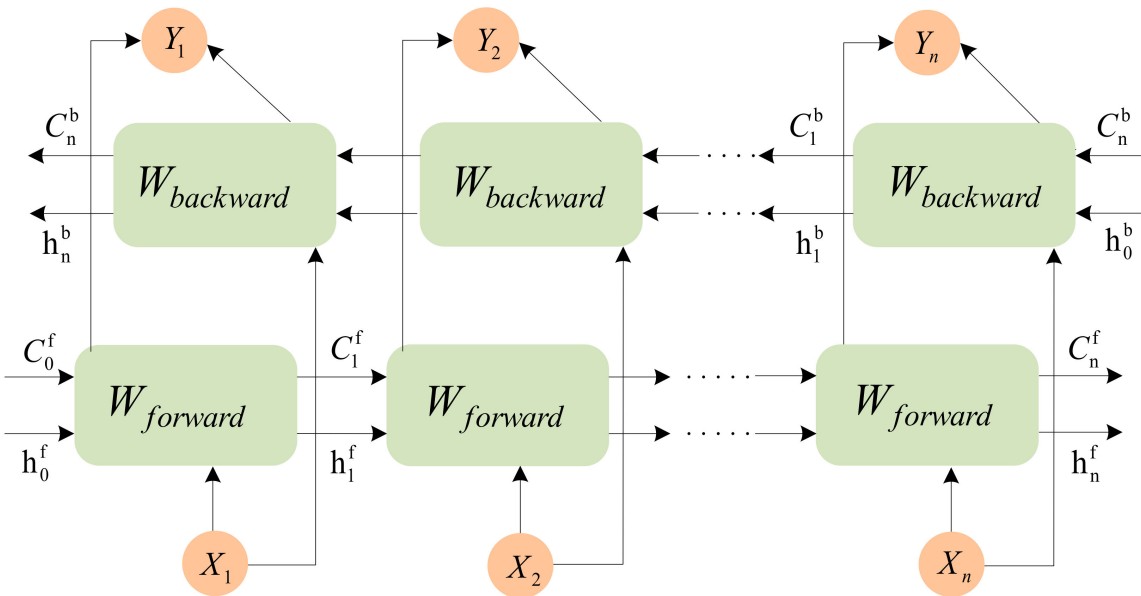

**Figure 4.** The schematic diagram of Bi-LSTM.

The Bi-LSTM relies on its unique recurrent hidden layer structure to model temporal dependency. The core of its recurrent hidden layer is a memory block containing memory units with self-connection, which is used to store the time state of the network at each time step [55]. The time state is controlled by three gates. The input gate is used to keep the memory content away from irrelevant inputs. The forget gate is used to forget useless information and the output gate is for transmitting outputs. The core unit of the Bi-LSTM is shown in Figure 5.

LSTM network can help to obtain local temporal information. However, in the traffic prediction problem, the characteristics of temporal information depend on more than adjacent time-slices. Due to the ability of the Transformer network to capture global temporal dependency, we input the features into the Transformer network to model global temporal dependencies.

The Transformer network includes a position embedding layer, multi-head self-attention layer, feedforward neural network layer, and batch normalization layer. In a Transformer, the self-attention mechanism based on the weighted average operation of the input features can flexibly and adaptively focus on different regions and capture more features. However, the traditional convolution operation uses an aggregation function on the local receptive field according to the convolutional weights, and they are shared in the whole feature map. Due to the different and complementary properties of self-attention and convolution, integrating these modules may benefit from the two paradigms. To enhance

the stability of multi-step prediction and reduce the accumulation of error, this paper attempts to add a $1 \times 1$ convolutional layer in front of the multi-head attention layer to extract intermediate features. Then, we further reuse and aggregate global temporal features by adding a $1 \times 1$ convolutional after the multi-head attention layer. The Conv-Transformer structure is shown in Figure 6. In Figure 6, X represents the input features, and Y represents the output features.

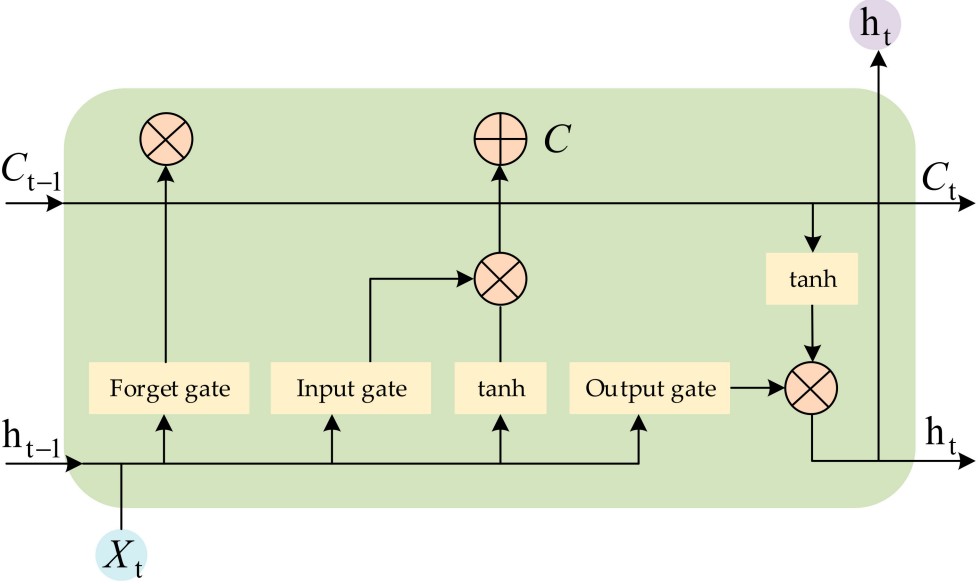

**Figure 5.** The core unit of the Bi-LSTM.

In the Transformer, the flow at a specific time step can be connected with all other flow through a self-attention mechanism. However, the attention mechanism in the Transformer layer treats different location nodes equally when capturing temporal features, which ignores the relative position relationship in the time series. Therefore, an embedded layer is needed to label the relative positions of different nodes. In this article, we use the embedding layer provided by PyTorch to embed positional relationships.

The core of the Transformer is the attention mechanism. The input of the attention mechanism includes queries, keys, and values vectors. In addition, the three vectors are obtained by multiplying the embedded feature $\hat{X}$ with the randomly initialized matrix $W^q$, $W^k$, and $W^v$ respectively, which is shown in Equation (5). The embedded feature $\hat{X}$ is obtained by the input data and position embedding layer. We further fuse and extract embedded features by CNN. Then, we calculate the dot product of given queries and keys. We continue to divide the dot product by the square difference of the dimension of keys. Then, the dot product operation is performed with values to learn two-way temporal dependency. Finally, we can obtain the attention scores of other positions through the function, which is shown in Equation (5):

$$
\begin{aligned}
Q &= \hat{X}W^q \\
K &= \hat{X}W^k \\
V &= \hat{X}W^v \\
Attention(Q, K, V) &= Softmax\left(\frac{QK^T}{\sqrt{d^k}}V\right),
\end{aligned}
\tag{5}
$$

where $Q$, $K$, and $V$ represent the queries, keys, and values for all the time points. $K^T$ represents the transpose matrix of $K$. $d^k$ represents the dimension of keys. We use the self-attention mechanism, $Q = K = V \in R^{T \times d_k}$. As for the multi-head attention used in this paper, $Q$, $K$, and $V$ are transformed through the linear layer according to the number of attention heads. Then the tensor dimension is reshaped. The tensors are calculated, respec-

tively, and then aggregated to obtain the final features. The process is shown in Equation (6):

$$MultiAttention(Q, K, V) = Concate(Softmax(\frac{Q_1 K_1^T}{\sqrt{d_1^k}} V_1), \dots Softmax(\frac{Q_n K_n^T}{\sqrt{d_n^k}} V_n))W^O, \tag{6}$$

where n is the number of attention heads, and $W^O$ is another linear output projection. $Q_n$, $V_n$, and $K_n$ are the queries, values, and keys obtained by the projection matrices of the n-th attention head. After extracting the features through the multi-head attention mechanism, the features that are further aggregated through the CNN are input into the feedforward neural network. Then, the global temporal characteristics can be obtained after being transmitted into the batch normalization layer through the feedforward neural network.

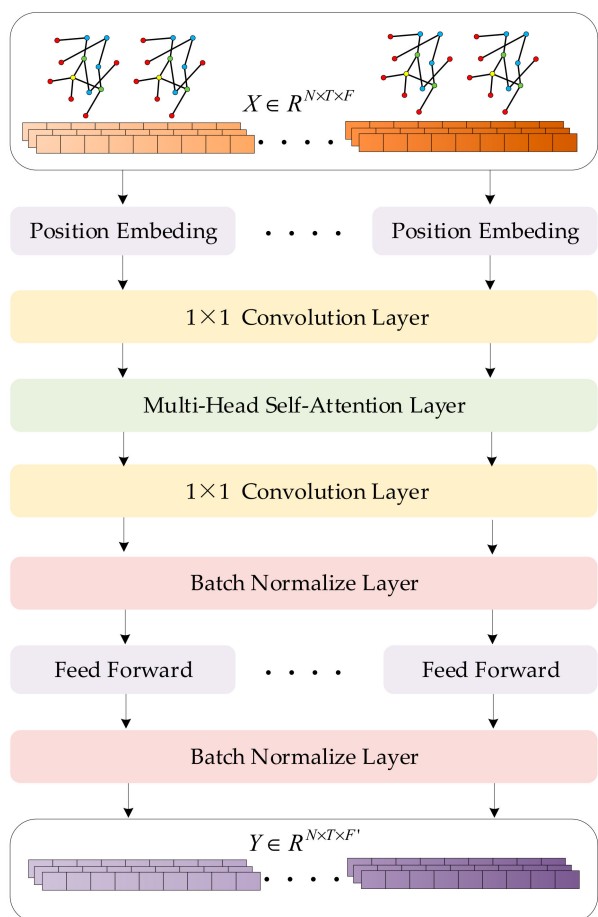

**Figure 6.** Structure diagram of Conv-Transformer.

### 3.4. Temporal Feature Fusion and Prediction Layer

The local temporal characteristics captured by the Bi-LSTM and the global temporal characteristics captured by Transformer are fused through a gate mechanism. The gate is shown in Equation (7).

$$g = sigmoid(f_l(Y_l) + f_g(Y_g)) \tag{7}$$

where $g$ is the gate to congregate the local and global temporal characteristics, *sigmoid*() is an activate function. $f_l$ and $f_g$ are the linear projections that transform the input data into a 1-D vector. $Y_l$ is the output of the Bi-LSTM, and $Y_g$ is the output of the Transformer. The

comprehensive temporal characteristics $Y_c$ are obtained by weighting $Y_l$ and $Y_g$ with gate $g$, which is shown in Equation (8).

$$Y_c = gY_l + (1 - g)Y_g \qquad (8)$$

After obtaining the comprehensive temporal characteristics of the first spatiotemporal block, we fuse the comprehensive temporal features with the obtained spatial features. In addition, we input the fused features into the next spatiotemporal block. After obtaining the final spatiotemporal features, we obtain the final prediction value through two CNN layers. We fuse and flatten the diffused high-dimensional features into the required feature dimension with the CNN. The high dimensional features are flattened to the inflow and outflow features. Meanwhile, the data are flattened to the time steps of the predicted target, which is shown in Equation (9).

$$Y = \mathrm{Conv}(\mathrm{Conv}(X)) \qquad (9)$$

Y and X are the output and the input of the GCTN, respectively.

## 4. Case Study

### 4.1. Data Sets and Evaluation Metrics

The subway passenger flow data used in this study are from the Smart Card Data (SCD) of the Shanghai Subway System, China. The study area and corresponding subway lines are shown in Figure 7. The field interpretation and examples of original data are shown in Table 1.

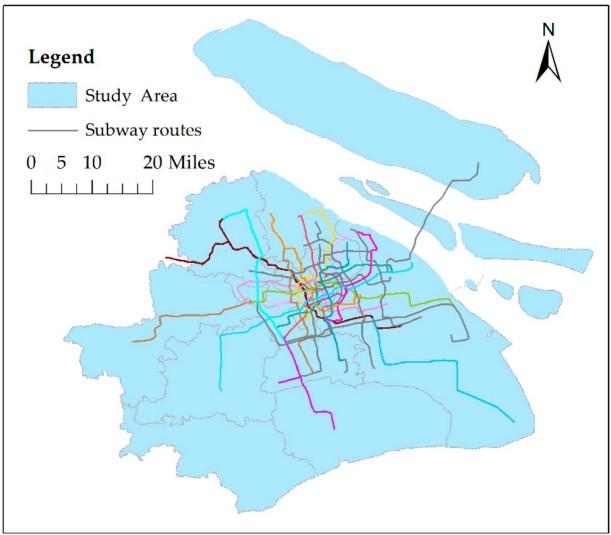

**Figure 7.** The study area and corresponding subway lines.

The data span is from 7 April to 30 April 2015. During this period, about 9 million cards were recorded every day, covering 289 subway stations. According to the people's public activity habits, this paper selects 5:30–23:00 as the research period. Besides, we take 10 min, 20 min, and 30 min as the time intervals for counting passenger flow. Take 10 min as an example, the data in 23 days can be divided into 2415 time-slices. The data are divided into a training set, a verification set, and a test set according to the ratio of 6:2:2. We use the average absolute error (MAE), root mean-square error (RMSE), and weighted average

absolute percentage error (WMAPE) for quantitative analysis and evaluating the prediction performance of different models. The evaluation metrics are shown in Equation (10).

$$\begin{aligned}
\text{MAE} &= \frac{1}{n} \sum_{i=1}^{n} |\hat{y}_i - y_i| \\
\text{RMSE} &= \sqrt{\frac{1}{n} \sum_{i=1}^{n} (\hat{y}_i - y_i)^2} \\
\text{WMAPE} &= \sum_{i=1}^{n} \left( \frac{y_i}{\sum_{j=1}^{n} y_j} \left| \frac{\hat{y}_i - y_i}{y_i} \right| \right)
\end{aligned} \tag{10}$$

where $\hat{y}_i$ is the predicted value, $y_i$ is the real value, $n$ is the total number of time-slices, and the $i$ is a specific time-slice.

**Table 1.** Data structure of the SCD and Example Data.

| Field | Description | Field Type | Example |
|---|---|---|---|
| CardNum | Unique number for each card | varchar | 602141128 |
| Date | Detailed date of transaction | datetime | 1 April 2015 |
| Time | Detailed date of transaction | datetime | 08:07:47 |
| LineName | Unique number of subway and name of subway station | varchar | No. 7 Shangda Road |
| Business | Travel way of trip | varchar | subway |
| Figure | Price of trip | float | 6.0 |
| Attribute | Discount or not | varchar | No discounts |

*4.2. Environment and Model Parameters*

The experiment runs on a GPU platform with an NVIDIA GeForce GTX3080 graphics card. We use several Python libraries to build the model, including scikit-learn and PyTorch.

After many experimental tests and parameter adjustments, the optimal parameters of the final model in the time intervals of 10 min, 20 min, and 30 min are similar. We take the dataset with 20 min Tis as an example. We employ the MSE as the loss function, and the calculation process is shown in Equation (11). We also used the early stopping technique to avoid overfitting. The model is trained for 1000 iterations, and it usually stops around 600 iterations. The number of spatiotemporal blocks is set as one, and the batch size is set as 45. The attention head is four, and the feature channels are set to 64. The number of GCN, LSTM, and Transformer layers is all set as one. The time intervals of the close, daily, and weekly patterns in the input data are 1:1:1. The input channel includes inflow and outflow. During the training process, the learning rate is $1 \times 10^{-3}$, and the Adam optimizer is adopted.

$$\text{MSE} = \frac{1}{n} \sum_{i=1}^{n} (y_i - \hat{y}_i)^2, \tag{11}$$

And then, we introduce the procedure of turning the model configurations, including the number of attention heads, the number of feature channels, and the order of Chebyshev polynomials, which are shown in Figure 8. We also study the effectiveness of the spatio-temporal blocks, which is shown in Table 2.

In Figure 8, the WMAPE changes little with various model configurations. The change in MAE is also relatively small. The RMSE has a more obvious change. When the attention head is less than four, the RMSE has a downward trend. When the attention head is larger than four, the MAE and the RMSE have an upward trend. In terms of feature channels, the MAE and RMSE both have a downward and then upward trend, and the turning point is 64. In Chebyshev orders, the RMSE has a downward and then upward trend, and the turning occurs when the order is close to three.

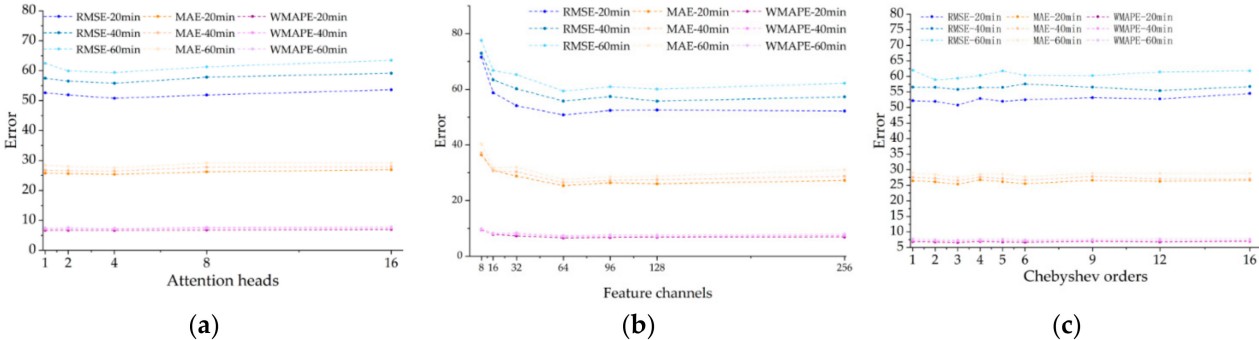

**Figure 8.** Evaluation metrics for 20 min TI obtained by GCTN with various model configurations. (**a**) Attention heads; (**b**) Feature channels; (**c**) Chebyshev orders.

**Table 2.** Evaluation metrics for 20 min TIs obtained by GCTN with spatiotemporal block.

| The Number of Spatiotemporal Blocks | MAE | RMSE | WMAPE |
|---|---|---|---|
| | 20/40/60 min | 20/40/60 min | 20/40/60 min |
| 1 | 25.35/26.47/27.54 | 50.79/55.78/59.38 | 6.60/7.03/7.38 |
| 2 | 26.28/26.90/28.08 | 51.55/55.12/58.54 | 6.73/6.99/7.40 |

In Table 2, we can find that the model with two spatiotemporal blocks improves the performance of RMSE. However, the performance of MAE has decreased. It means that the prediction performance decreases in the period of low passenger flow, but the ability to capture the peak value is further improved. The peak value is more critical for subway dispatching. In consequence, the superposition of spatiotemporal blocks is helpful.

Meanwhile, in order to explore the subway passenger flow prediction by different temporal combination modes, we constructed 11 different temporal combination mode datasets and tested them. Where H represents the close patterns, D represents the daily patterns, and W represents the weekly patterns. The results of RMSE are shown in Figure 9.

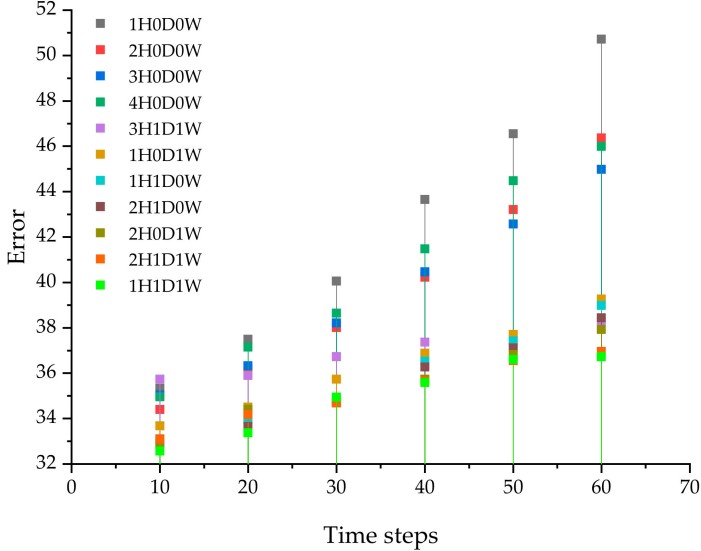

**Figure 9.** Prediction results of GCTN in different time patterns data sets—RMSE.

In Figure 9, we can find that the prediction performance is the worst when there is only the input data of close patterns. We attempt four kinds of input data involving only close patterns. With the increase in close patterns, the prediction performance is improved. However, with the increase in the number of close patterns in the dataset,

the prediction performance will not always get better. The prediction performance is significantly improved when the close pattern is combined with the daily pattern or weekly pattern. When the three kinds of patterns are combined, the prediction performance is the best. The best overall prediction performance is the 1HD1W. The prediction performance of 2H1D1W is also good, sometimes even better than 1H1D1W. However, on the whole, the RMSE of 2H1D1W is relatively high. In terms of 3H1D1W, the RMSE has the most stable change in multiple time steps, which means the prediction stability is good. However, the RMSE of 3H1D1W is generally higher than those of 1H1D1W and 2H1D1W.

### 4.3. Baseline Models

To make a comparison, we compare the proposed model with nine baseline models (including one statistical method, three machine learning methods, and seven deep learning methods) based on the same combination of input patterns. The baselines are briefly described as follows:

- Linear Regression (LR) [56]: A statistical model, which can perform for future predictions based on the linear relationship between the input data and the target data.
- K-Nearest Neighbor (KNN) [19]: A machine learning method in which the passenger flow of the target node is calculated according to the average passenger flow of its nearest neighbor.
- Support Vector Regression with Radial-basis Function (RSVR) [15]: A typical machine learning method. The radial basis function is the kernel of SVR in scikit-learn.
- Support Vector Regression with Linear Function (LSVR) [16]: A typical machine learning method. The linear function is the kernel of SVR in scikit-learn.
- Bidirectional Long Short-Term Memory (Bi-LSTM) [42]: A deep learning model composed of two layers of LSTM, which can capture both past and future information.
- Transformer [53]: A deep learning model. It is based on a self-attention mechanism and can model global temporal dependency.
- Convolutional Neural Network (CNN) [38]: A typical deep learning model. In traffic flow prediction, the data are processed into a two-dimensional image format as input of the neural network. In addition, the vertical axis represents the subway station, and the horizontal axis represents the time.
- Spatial-Temporal Graph Convention Network (STGCN) [44]: The model captures spatial dependencies using GCN and captures temporal features using 1D convolution combined with Gated Linear Unit (GLU).
- Spatial-Temporal Transformer Network (STNN) [53]: The model creates a spatiotemporal Transformer mode, which captures temporal features by temporal Transformer, and captures spatial features by combining spatial Transformer and GCN.
- GCTN_NoLSTM: We deleted the LSTM component in the spatiotemporal block of the GCTN.
- GCTN_NoTransformer: We deleted the Transformer component in the spatiotemporal block of the GCTN.

### 4.4. Results and Discussion

We analyzed and discussed the multi-step prediction results with 10 min, 20 min, and 30 min time intervals. At the same time, we compared the predicted results with the real values in some specific stations. We analyzed the prediction performance in the period that the subway passenger flow changed quickly. We showed the prediction performance of the model during the working day. We explored the influence of the combinations of time patterns. Finally, the performance of the improved components in the GCTN is discussed.

#### 4.4.1. Comparison of Different Model Prediction Results

We use the same data and multi-step prediction mode in different models. The features of passenger flow are diffused by a $1 \times 1$ convolutional layer. In addition, the prediction results of the deep learning model are finally aggregated by two $1 \times 1$ convolutional layers.

The prediction evaluations for different models in different time intervals are shown in Tables 3 and 4.

**Table 3.** Evaluation results of different models and TIs (10 min).

| Model | MAE | RMSE | WMAPE |
|---|---|---|---|
| | 10/20/30/60 min | 10/20/30/60 min | 10/20/30/60 min |
| LR | 20.54/21.18/21.74/23.89 | 43.32/46.13/47.11/52.36 | 11.65/12.28/12.61/14.00 |
| KNN | 17.22/17.58/17.94/19.27 | 35.39/36.59/38.19/43.61 | 9.36/9.60/9.85/10.74 |
| RSVR | 19.22/19.52/19.86/21.02 | 41.40/42.32/43.16/46.11 | 9.70/9.90/10.13/10.98 |
| LSVR | 19.56/20.00/20.16/20.93 | 37.87/38.61/38.80/40.64 | 10.28/10.50/10.49/10.88 |
| Bi-LSTM | 18.09/18.34/18.67/20.07 | 34.74/35.71/36.66/40.55 | 9.48/9.64/9.84/10.75 |
| Transformer | 18.53/18.92/19.07/20.45 | 35.96/37.11/37.72/41.56 | 9.86/10.05/10.20/11.11 |
| CNN | 22.35/22.14/22.12/23.85 | 43.04/42.92/42.20/45.76 | 11.90/12.00/11.80/12.78 |
| STGCN | 17.39/17.26/17.34/18.21 | 34.86/34.78/35.38/37.21 | 9.26/9.19/9.26/9.74 |
| STTN | 17.29/17.31/17.59/18.63 | 33.43/33.58/34.33/37.02 | 8.92/8.95/9.13/9.85 |
| GCTN_NoLSTM | 17.17/17.41/17.81/18.83 | 33.63/34.51/35.85/38.92 | 9.14/9.20/9.54/10.14 |
| GCTN_NoTransformer | 17.26/17.32/17.50/18.55 | 33.61/34.24/34.98/38.25 | 9.01/9.16/9.30/9.96 |
| GCTN | 17.26/17.42/17.75/18.65 | 32.57/33.37/34.95/36.73 | 8.86/8.98/9.24/9.81 |

**Table 4.** Evaluation results of different models and TIs (20 min and 30 min).

| Model | 20 min TI | | | 30 min TI | | |
|---|---|---|---|---|---|---|
| | MAE | RMSE | WMAPE | MAE | RMSE | WMAPE |
| | 20/40/60 min | 20/40/60 min | 20/40/60 min | 30/60 min | 30/60 min | 30/60 min |
| LR | 33.82/36.64/43.05 | 77.58/86.94/102.96 | 10.19/11.19/13.39 | 45.81/49.81 | 112.13/127.41 | 9.67/10.45 |
| KNN | 27.06/28.74/30.68 | 60.18/67.39/75.80 | 7.58/8.13/8.77 | 37.05/41.39 | 88.47/107.59 | 7.19/8.12 |
| RSVR | 32.78/33.99/35.74 | 82.15/85.94/89.87 | 8.18/8.63/9.23 | 47.56/51.15 | 131.40/141.20 | 8.01/8.89 |
| LSVR | 33.02/33.99/35.74 | 63.54/66.58/67.92 | 8.40/8.61/8.70 | 46.56/48.80 | 89.55/95.31 | 7.77/7.95 |
| Bi-LSTM | 27.09/28.50/30.23 | 55.27/60.92/66.83 | 7.17/7.66/8.29 | 38.67/42.10 | 80.49/94.44 | 6.86/7.61 |
| Transformer | 29.43/30.65/32.10 | 58.40/62.88/68.44 | 7.78/8.14/8.80 | 40.34/44.07 | 83.71/97.68 | 7.20/8.13 |
| CNN | 37.38/39.17/41.47 | 73.64/75.45/81.11 | 10.05/10.36/10.88 | 49.80/56.16 | 99.15/115.57 | 8.91/10.34 |
| STGCN | 26.96/27.66/29.14 | 56.56/58.97/64.74 | 7.22/7.41/7.82 | 36.76/39.93 | 78.48/89.09 | 6.44/7.06 |
| STTN | 27.86/27.40/28.39 | 53.21/56.96/60.49 | 6.93/7.20/7.55 | 38.25/40.93 | 73.92/85.63 | 6.23/7.11 |
| GCTN_NoLSTM | 27.16/28.03/29.36 | 54.11/58.66/62.67 | 7.02/7.42/7.86 | 37.01/39.10 | 74.80/85.18 | 6.41/6.98 |
| GCTN_NoTransformer | 26.81/28.25/29.70 | 53.48/58.21/62.55 | 6.96/7.35/7.75 | 35.07/**37.55** | 73.81/83.22 | 6.15/6.74 |
| GCTN | 25.35/26.47/27.54 | 50.79/55.78/59.38 | 6.60/7.03/7.38 | 34.68/37.61 | 70.72/81.35 | 5.99/6.62 |

By analyzing MAE and WMAPE, we can find that the GCTN can achieve better results on subway passenger flow datasets with different time intervals. However, MAE and WMAPE for the GCTN are not always better than those of other models. For example, in 10 min time interval, the MAE of the STGCN is better than that of the GCTN. In addition, it may be that the average level of subway flow in the 10 min time interval is low, which results in a small difference between low flow and high flow. In addition, in the 20 min and 30 min time intervals, the MAE of the GCTN is better because of the further expansion of the difference between the high and low flow.

By analyzing RMSE, it can be seen that the results in the machine learning methods are generally better than those in the statistical method in 10 min time interval. However, in the 20 min and 30 min time intervals, the results in the statistical method LR are better than those in the RSVR, which is a machine learning method. The reason may be the reduction in the number of samples. Compared with the statistical method and machine learning methods, most deep learning methods have a better prediction performance. Firstly, the LSTM and Transformer models considering temporal characteristics perform better than the statistical method and machine learning methods. Moreover, the prediction performance of the Bi-LSTM capturing local temporal characteristics is better than the Transformer capturing global temporal characteristics. It may be that the local temporal characteristics are easier to capture than the global temporal characteristics in subway passenger flow. Secondly, the STGCN and STTN can capture spatiotemporal features. The STGCN captures spatiotemporal features, respectively, through graph convolutional networks. Compared with the STGCN, the STTN can capture fixed and dynamic spatial features with the GCN

and Transformer. In addition, it further integrates global temporal features. Therefore, the STGCN and STTN perform better than the Bi-LSTM and Transformer. Likewise, the prediction performance of the STTN is better than that of the STGCN. Lastly, the GCTN fuses local and global temporal features and improves the influence of the global stations in the GCN, which has better prediction results than the STTN and STGCN. To further compare the prediction performance of the fusion of local temporal features and global temporal features with the single temporal model, we modified the complete GCTN model by deleting some components in the GCTN. In this way, we get two new models, the GCTN_NoLSTM and GCTN_NoTransformer. At different time intervals, we find that the GCTN has a better prediction performance than the GCTN_NoLSTM and GCTN_NoTransformer. The results show that the prediction performance of comprehensive temporal characteristics is better than that of single temporal characteristics.

Taking the 20 min time interval as an example, we calculate the prediction evaluation by predicting the passenger flow of 20 min, 40 min, and 60 min after the current time step. In terms of RMSE, compared with statistical method and machine learning methods, the prediction performance of the GCTN is significantly improved. In addition, it increases with the increase in the range of time steps. In addition, in multi-step prediction, the improvement intervals of the GCTN are 15.60~38.17%/16.22~35.84%/12.57~42.33%. In deep learning methods, compared with the CNN, the improvement performance of the GCTN is 31.03%/26.07%/26.79%. Compared with the Bi-LSTM and Transformer, which model the temporal dependency, the improvement performance of the GCTN is 8.11%/8.44%/11.15% and 13.03%/11.29%/13.24%, respectively. Compared with the STGCN and STTN which can model spatiotemporal features, the improvement performance of the GCTN are 10.20%/5.41%/8.28% and 4.55%/2.07%/1.84%, respectively.

### 4.4.2. Ablation Studies

In order to explore the effectiveness of the improvements in the adjacent matrix and model, we conducted the following comparative experiments. The error results are shown in Table 5.

**Table 5.** Prediction performance of GCTN model and the model with ablation studies. The time interval is 10 min.

| Model | MAE | RMSE | WMAPE |
| | 10/20/30/60 min | 10/20/30/60 min | 10/20/30/60 min |
|---|---|---|---|
| GCTN_1 | 17.20/17.48/17.67/18.62 | 33.39/34.61/35.39/37.77 | 9.04/9.24/9.33/9.91 |
| GCTN_2 | 17.69/17.91/18.02/18.97 | 34.58/35.48/35.99/38.67 | 9.18/9.32/9.45/10.08 |
| GCTN_3 | 17.39/17.59/17.81/18.66 | 33.57/34.64/35.38/37.73 | 9.05/9.24/9.38/9.94 |
| GCTN_4 | 18.44/18.64/19.27/19.74 | 34.94/35.91/37.06/39.33 | 9.29/9.48/9.69/10.24 |
| GCTN | 17.26/17.42/17.75/18.65 | 32.57/33.37/34.95/36.73 | 8.86/8.98/9.24/9.81 |

1.  GCTN_1: the GCTN model using the un-improved adjacent matrix;
2.  GCTN_2: the GCTN model uses the improved adjacent matrix. The improved Transformer is used, but the dot product attempt of the Laplacian operator in the GCN is removed;
3.  GCTN_3: the GCTN model uses the improved adjacent matrix. The improvement in the GCN is used, but the convolutional layer added in a Transformer is removed;
4.  GCTN_4: there is an original GCTN model with no improvement. In other words, the improvements in the adjacent matrix, GCN, and Transformer are removed.

In Table 5, compared with the GCTN_4, the GCTN has a better prediction performance. It shows that our improvement in the adjacent matrix, the structure of the Transformer, and the GCN are effective and significant.

The GCTN_1 is a comparative model to verify the effectiveness of the improved adjacent matrix. Its basic architecture is the same as that of the GCTN. The results show that the improved adjacent matrix is helpful in prediction. The MAE of GCTN_1 is lower

than GCTN in some time steps. However, the RMSE and WMAPE are higher than the GCTN in each time step. The results indicate that for the subway network, the spatial influence of the import and export of the subway station exists and is worth analyzing.

The GCTN_2 removed the improvement of the GCN. Its prediction performance is only better than the GCTN_4. However, the MAE, RMSE, and WMAPE of the GCTN_2 are obviously higher than those of the GCTN. The results show that spatial influence is very important for subway passenger flow prediction, and it is necessary to enhance the influence of global stations on specific stations.

The GCTN_3 removed the CNN layer in the improved Transformer structure so that the extraction steps of intermediate features are reduced. The prediction performance of the GCTN_3 shows that the CNN layer is beneficial for extracting intermediate features and aggregating the global temporal features, which are captured by the self-attention mechanism.

### 4.4.3. Prediction Results of Specific Subway Stations

According to the evaluation results of different models, we select some deep learning models (CNN, Bi-LSTM, Transformer, STGCN, STTN, GCTN) for further discussion. Figures 10 and 11 show the real passenger flow and corresponding predicted flow at the Xujiahui Station and the People's Square Station, including half a non-working day and four working days. In previous studies [28], the fluctuation range of passenger flow on weekends is relatively small, which results in little demand for subway scheduling. Therefore, we focus on the passenger flow in four working days, and then discuss the details.

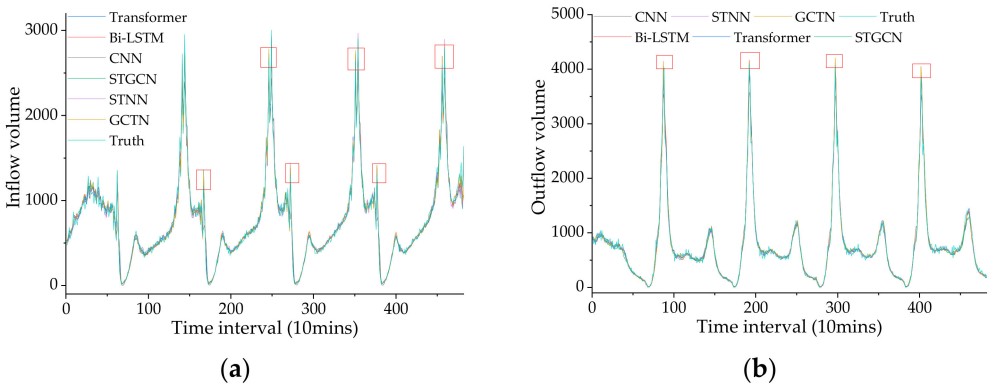

(**a**)        (**b**)

**Figure 10.** The prediction of passenger flow in the Xujiahui Station, (**a**) Inflow volume; (**b**) Outflow volume. The time interval is 10 min and the test period covers half a non-working day and four working days.

In Figures 10 and 11, we can learn the characteristics of the subway passenger flow. For example, the peak values of the passenger inflow and outflow on weekdays are about 6 p.m. and 9 a.m., respectively, which corresponds to the off-duty and on-duty hours. Meanwhile, we can see that the GCTN model has relatively good prediction results for the peak value, especially in the inflow prediction. The GCTN can predict well in almost every peak value, which is more beneficial for the reference of subway dispatching. Meanwhile, we show the long-term predictions of three models (GCTN, STGCN, STTN) that have a better prediction performance. We draw the results of multi-step prediction to the same time step and observe the difference in multi-step prediction performance in peak fitting. The results are shown in Figure 12. We can find that the prediction performance of each time step is relatively good for the peak fitting, and the predicted flow is often larger than the real flow. It is helpful for subway dispatching.

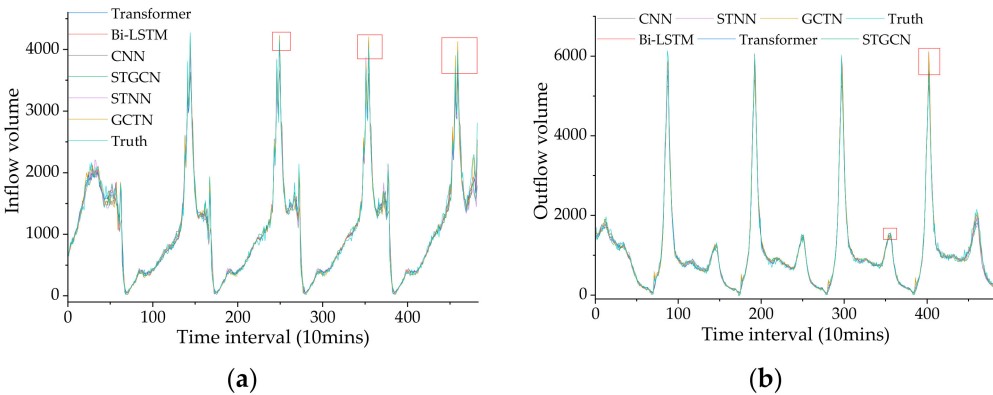

**Figure 11.** The prediction of passenger flow in the People's Square Station, (**a**) Inflow volume; (**b**) Outflow volume. The time interval is 10 min and the test period covers a half non-working day and four working days.

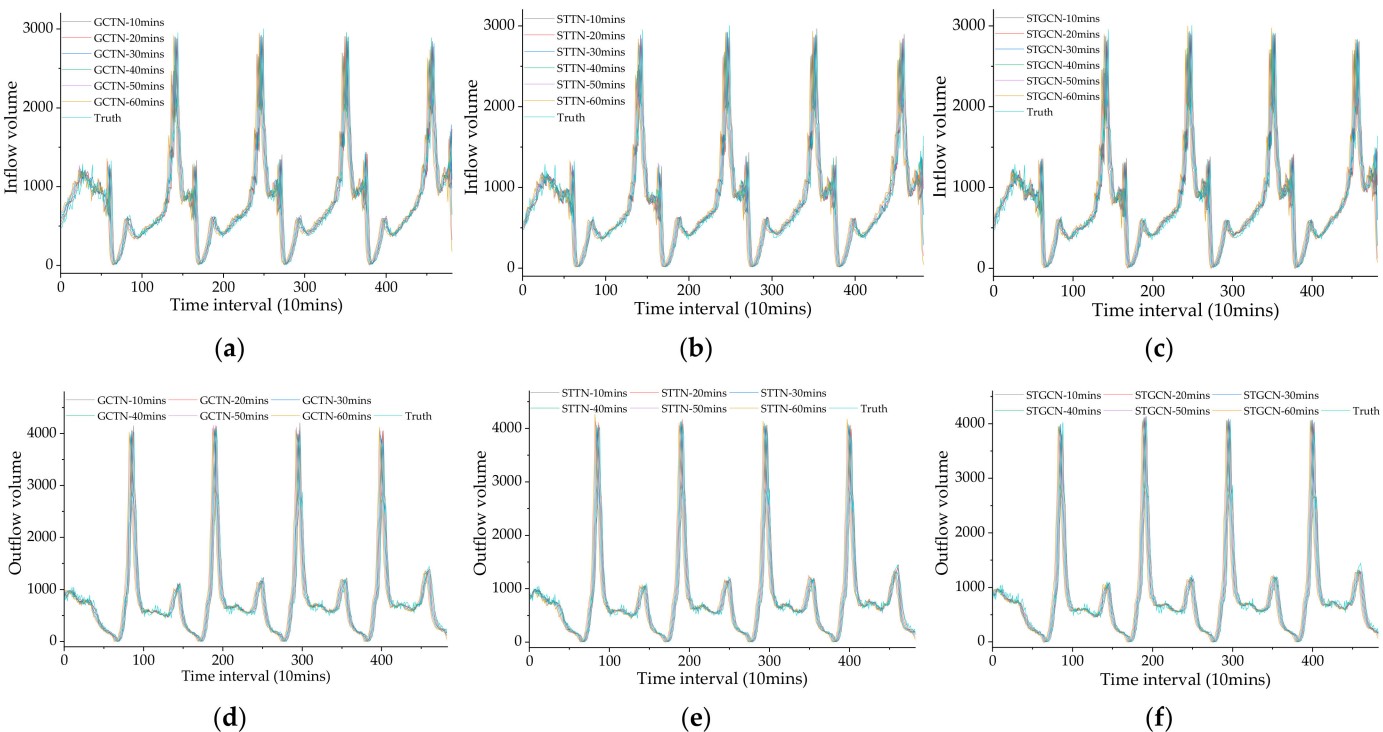

**Figure 12.** The visualization of long-term prediction with inflow volume and outflow volume in the Xujiahui Station of which time interval is 10 min, (**a**) GCTN inflow volume; (**b**) STTN inflow volume; (**c**) STGCN inflow volume; (**d**) GCTN outflow volume; (**e**) STTN outflow volume; (**f**) STGCN outflow volume.

As we all know, subway dispatching may become difficult when the passenger flow changes quickly. Therefore, we need to analyze the evaluation metrics during the period when passenger flow changes quickly. In a day, for the inflow volume, there are three periods that the passenger flow changes quickly, namely, 5:30 a.m. to 6:30 a.m., 5:30 p.m. to 8:00 p.m., and 9:30 p.m. to 11:00 p.m. For the outflow volume, the periods that the passenger flow changes quickly are 8:00 a.m. to 11:00 a.m. and 6:00 p.m. to 8:00 p.m. Therefore, to further prove the practicality of the model in subway passenger flow prediction, this paper attempts to divide the period that passenger flow changes quickly. The results are shown in the rectangular box in Figures 13 and 14. In addition, we calculate the RMSE of the three models in this period, which is shown in Table 6.

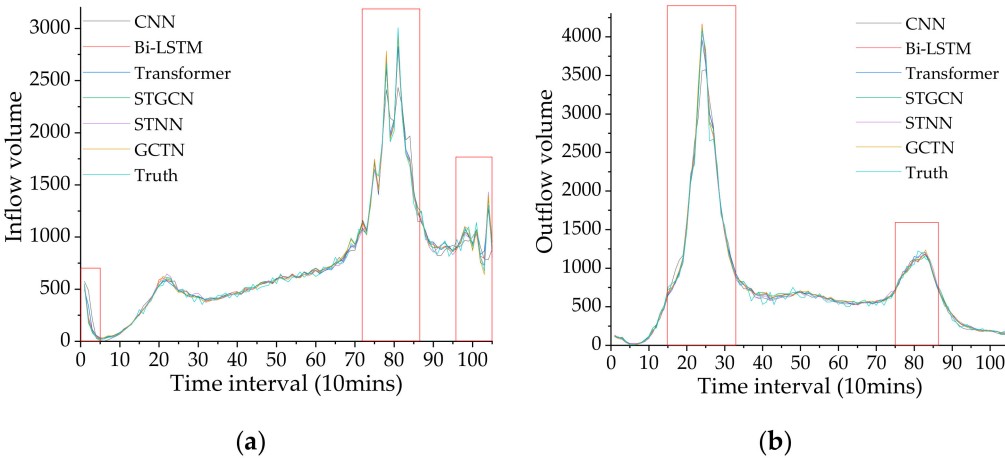

**Figure 13.** The time period that the passenger flow changes quickly in the Xujiahui Station. (**a**) Inflow volume; (**b**) Outflow volume.

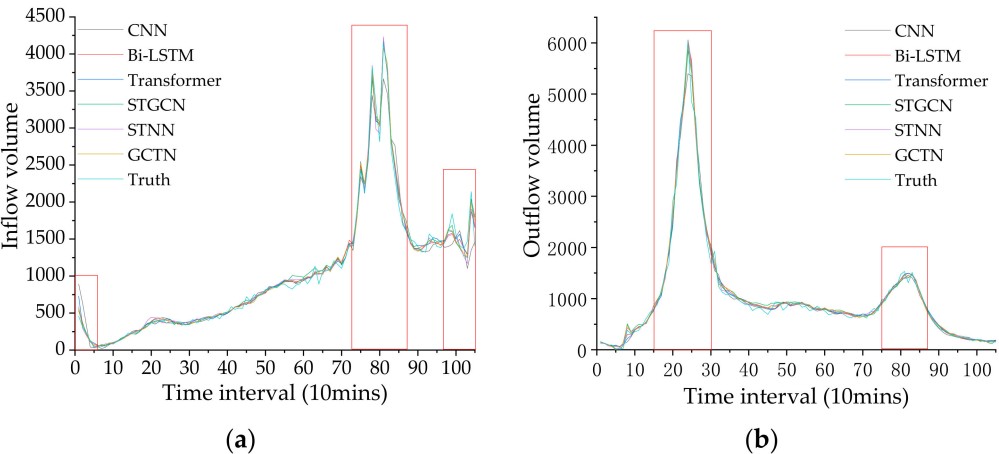

**Figure 14.** The time period that the passenger flow changes quickly in the People's Square Station. (**a**) Inflow volume; (**b**) Outflow volume.

**Table 6.** The prediction performance for the period when the passenger flow changes quickly at the whole station—RMSE.

| Model | Inflow Volume | Outflow Volume |
|---|---|---|
| | 10/20/30/60 min | 10/20/30/60 min |
| STGCN | 34.30/34.32/33.47/33.89 | 63.19/63.16/64.60/65.79 |
| STTN | 34.20/34.63/34.69/35.91 | 60.44/61.07/62.57/64.57 |
| GCTN | 33.33/34.65/34.33/34.35 | 60.05/61.63/65.51/65.70 |

In Figures 13 and 14, we can see that the period of subway passenger flow changes quickly at the two stations is the same. This paper compares the long-term prediction performance of the GCTN, STTN, and STGCN during the period when passenger flow changes quickly at the whole station. In Table 6, the GCTN has a better long-term prediction performance in the period when passenger flow changes quickly, which is better than the STTN in the inflow, and better than the STGCN in the outflow. In general, during the period when passenger flow changes quickly, the prediction performance of the GCTN on subway passenger flow is better than the relevant models listed in this paper.

To further study the stability of model prediction and the prediction performance of different periods in a day, we divide a working day into 17 h. Considering all working

days in the test set, we calculate the related error of these 17 h. In this way, we can explore the prediction performance of models in different periods more specifically. As the analysis basis, the evaluation metrics (MAE, RMSE, WMAPE) are calculated every hour. The results are shown in Figures 15 and 16.

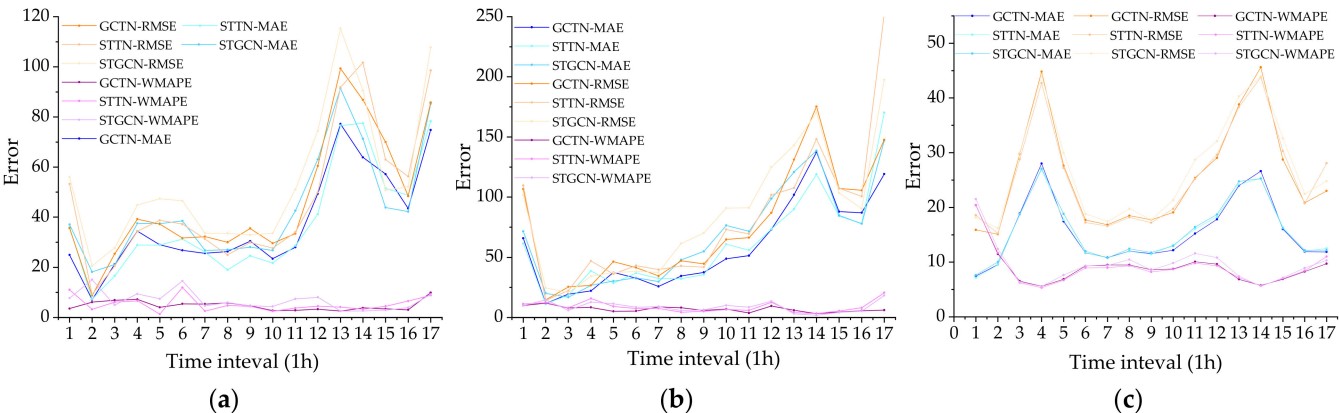

**Figure 15.** Inflow prediction performance of different times in the working day, with each time interval of 60 min. (**a**) Xujiahui station; (**b**) People's Square station; (**c**) All stations.

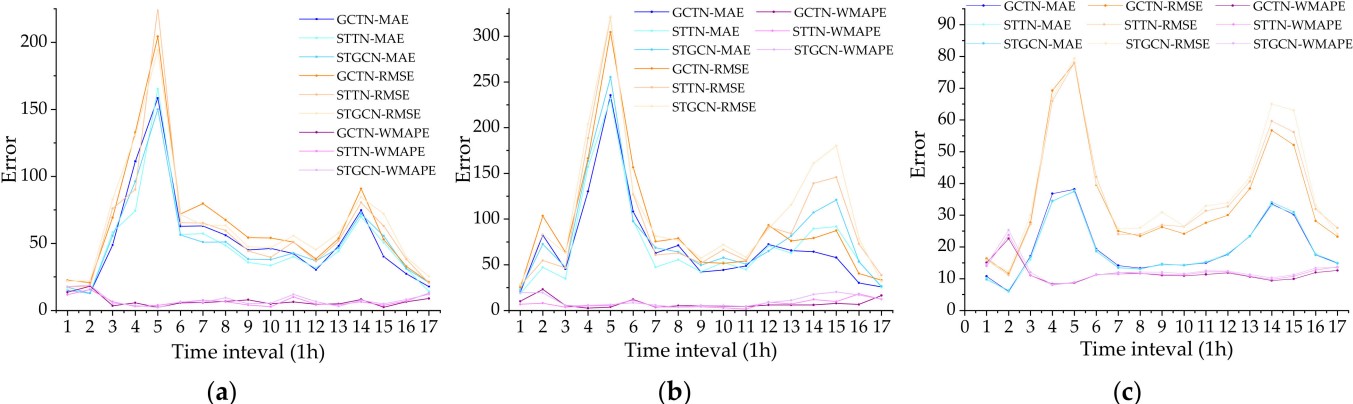

**Figure 16.** Outflow prediction performance of different times in the working day, with each time interval of 60 min. (**a**) Xujiahui station; (**b**) People's Square station; (**c**) All stations.

In Figures 15 and 16, we can see that the prediction performance of the proposed model changes over time in the working day. For the passenger inflow and outflow, the prediction performance of the model changes in twists and turns. Corresponding to the period when flows change quickly at Xujiahui Station and the People's Square Station in Figures 13 and 14, it can be seen that the prediction error increases sharply. The reason may be that the data fluctuates greatly and has strong randomness. It is difficult to capture the spatiotemporal characteristics. In consequence, the prediction performance of models is to be improved during the periods when the passenger flow changes quickly.

We make an analysis for the prediction performance in a working day of these models. Compared with the STGCN and STTN, the evaluation metrics of the GCTN change relatively little in the working day. We can find the RMSE of the GCTN is better in the inflow of Xujiahui station and the outflow of the People's Square Station. However, in all stations, we can find that the RMSE of the GCTN is better in outflow, and the STTN is better in inflow. It shows that the GCTN has different performances in different sites, and the overall prediction performance is more stable in the outflow of the subway.

### 4.4.4. The Spatial Influence of the Surroundings about Stations

To study the spatial factors affecting the prediction of the subway passenger flow, we analyzed the import and export of subway stations, the point of interest (POI) around subway stations, and the RMSE of the prediction of subway passenger flow. Specifically, these elements include import and export quantity at the station, the number and type of POI within 200 m of the station, and the RMSE of the specific station.

We used K-means clustering to classify these elements into three categories, respectively. In addition, the heat maps of clustering results are drawn to observe the spatial distribution of passenger flow prediction errors and the influence of POI and import and export on passenger flow prediction. The results are shown in Figure 17.

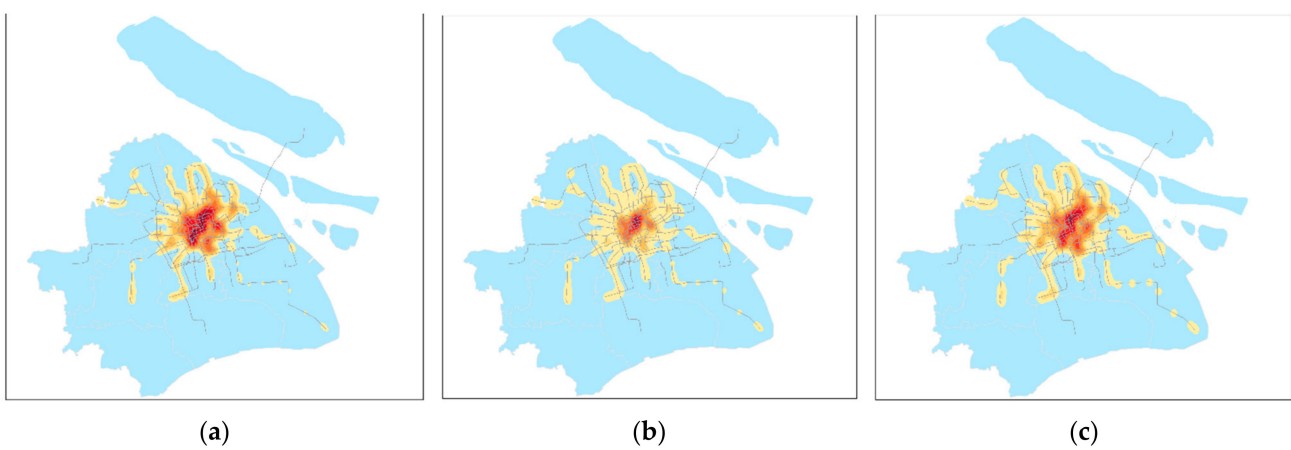

(**a**)          (**b**)          (**c**)

**Figure 17.** Heat maps of spatial factors and RMSE. The darker the color, the higher the value. (**a**) The count of import and export; (**b**) The number and type of POI; (**c**) The mean RMSE on stations.

The subway stations with higher RMSE are concentrated, and most of them are located in the city center. In addition, subway stations with more imports and exports or more POI are also concentrated in the city center. Therefore, we believe that the number and type of POI and import and export quantity will influence the accuracy of the flow prediction at the subway station. In addition, it is positively correlated with the RMSE. The more import and export, the higher the RMSE of flow prediction. The less the number and types of POI around the subway station, the lower the RMSE of the flow prediction on the subway station. It proves the correctness of bringing import and export into the spatial influencing factors of the subway network, and the correctness of our improved adjacent matrix to combine import and export quantity as part of the spatial features.

### 5. Conclusions

Aiming at the problem of subway passenger flow prediction, we argue that the existing works ignore the spatial influence of the import and export of subway stations and the influence of global stations on specific stations. In addition, many methods are based on a single RNN model or its variations, or the Transformer model, which has limitations in capturing temporal features. This paper proposed a hybrid neural network, GCTN, to solve these problems.

We use Shanghai subway passenger flow data to test. The results show that MAE, RMSE, and WMAPE have achieved good performance in multi-step prediction. The GCTN has a better prediction performance in capturing the peak and the period when passenger flow changes quickly, which is more conducive to the practical application of the model. We compared the effects of different temporal combination modes, which show the combination of the close, daily, and weekly patterns can improve the prediction performance. Meanwhile, we verified the proposed improvements in the GCTN, and we think the combination of the CNN and Transformer is helpful. We found that the RMSE of

the predicted flow of the station is positively correlated with import and export quantity and POI quantity.

However, some limitations still exist. Firstly, the period of the validation data set is not long enough to study the influence of long-temporal factors, such as seasons. Secondly, the influence of external factors, such as weather is not considered. Finally, we do not study the dynamic spatial characteristics, which may improve the spatial dependency. In the future, we will further explore the influence of external features and study the application of the GCTN in longer data sets. We also intend to study the influence of dynamic spatiotemporal features in the deep learning model and the differences between different kinds of attention mechanisms in global temporal features.

**Author Contributions:** Conceptualization, Yong Han; Data curation, Zhihao Zhang; Formal analysis, Zhihao Zhang and Tongxin Peng; Funding acquisition, Yong Han; Investigation, Zhihao Zhang; Methodology, Zhihao Zhang and Tongxin Peng; Project administration, Yong Han; Resources, Yong Han; Software, Zhihao Zhang; Supervision, Ge Chen; Validation, Zhihao Zhang; Visualization, Zhihao Zhang and Zhenxin Li; Writing—original draft, Zhihao Zhang; Writing—review & editing, Zhihao Zhang and Tongxin Peng. All authors have read and agreed to the published version of the manuscript.

**Funding:** This research was funded by the Natural Science Foundation of Shandong Province, China (Grant No. ZR2020MD020).

**Institutional Review Board Statement:** Not applicable.

**Informed Consent Statement:** Not applicable.

**Data Availability Statement:** Data sharing is not applicable to this article.

**Acknowledgments:** Thanks for the data provided by Shanghai Public Transportation Group.

**Conflicts of Interest:** The authors declare no conflict of interest.

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
