# Peer review of "A Comprehensive Spatio-Temporal Model for Subway Passenger Flow Prediction"

_ijgi, doi:10.3390/ijgi11060341_

Round 1

Reviewer 1 Report

  1. Lines 94 to 100. The original characteristics and the characteristics captured by GCN. The latter is spatial characteristics, what are the original characteristics? Why is fusion a spatiotemporal feature? It doesn't seem to express clearly.
  2. You mentioned the Transformer model and make an improvement on it, but there aren’t any references about the paper. (As I know, the paper is ATTENTION IS ALL YOU NEED. )
  3. In Line 280-283, “The GCN model captures spatial features. The Bi-LSTM model captures local temporal features.” But the reasons for this are not found in the paper. For example, why is Bi-LSTM able to extract the local temporal characteristics and why is Transformer able to extract the global temporal characteristics? What is the difference between them?
  4. The analysis of the experimental part is too long, and the results that can be seen immediately on the table have been repeatedly explained. It is recommend that the analysis of important results should be highlighted.
  5. It is mentioned that the existing prediction models are mostly the short-term prediction models, and the proposed model can predict long-term subway passenger flow. However, the comparative experimental proof on this is not been provided. And the longest term of the experiments is 60 minutes, which just can be seem as a short-term prediction.
  6. There are many redundant fragments, unclear contents in the paper.
  7. The language should be improved.

Reviewer 2 Report

Nothing to tell on this manuscript which is interesting, although difficult in some parts to read due to too many abbreviations. However, it misses the central point: for a journal called IGJI, modelling is reported as per se, and results in terms of spatial implications should be clearly stated as land use around the station, access costs, etc. (moreover, additional outcomes in terms of environmental benefits, energy savings, etc. would enrich the spatial outcomes analysis); at least for the case study described. None of the above is reported in the manuscript, which results unfit for publication in the present form". I think that the paper, in the present form would be better fit the Future Transportation serie

Reviewer 3 Report

The present study focuses on an accurate prediction of subway passenger flow by using a hybrid neural network model. The paper is written in a readable level of English, nevertheless, there are some grammar errors and language problems that should be fixed.

The paper is interesting and methodological reasoning is presented brilliantly.

I suggest removing section 2 to diminish the oversize of the article, and strengthening section 1 by using some info from the removed section as well I suggest the reorganization of section 1 for better traceability and intelligibility of the paper.

Some subsections for the Introduction must be presented such as (four subsections would be useful) background, literature review, methodological reasoning, the aim of the study & paper organization. The role and importance of urban railways in urban sustainable development could be elaborated better in the present background of the introduction at the beginning (e.g. https://doi.org/10.3390/su131810158).

Sections 5 to 7 could be presented in a more holistic and compact way, the current version is hard to follow up for readers. The current conclusions are like a summary of the present work, leave that part brief please, instead of this focus on presenting the fundamental gain of the study for the related literature. Please be more specific and informative about the results of the study in the abstract.

Round 2

Reviewer 1 Report

The authors reviesed the paper carefully according to the reviewers' comments.  There are still some minor suggestions to authors.

1. There seems to be something wrong with the picture on line 308.

2.  Would it be better to change the title of your fourth chapter to be Experiment ?

Reviewer 2 Report

The authors have met parts of the requirements, yet the manuscript does not fit entirely with the journal scope.